# INTEGRAL PINNS FOR HYPERBOLIC CONSERVATION LAWS

**Manvendra P. Rajvanshi & David I. Ketcheson**[*]
Applied Mathematics and Computational Science
King Abdullah University of Science and Technology
Thuwal 23955-6900, Kingdom of Saudi Arabia
{manvendra.rajvanshi,david.ketcheson}@kaust.edu.sa

## ABSTRACT

Traditional physics-informed neural networks (PINNs) are trained based on differential equations and thus have difficulty capturing shock discontinuities in weak solutions of hyperbolic PDEs, since the differential equation doesn't apply at the discontinuity. We propose Integral PINNs (IPINNs), which are trained based on the integral form of the conservation law, which holds at both continuous and discontinuous points of the solution. We use neural nets to model the integrals of the solution instead of the solution itself. We apply IPINNs to systems of hyperbolic conservation laws and show that they are much better at capturing the correct location and speed of shocks, compared to traditional PINNs. We also present a heuristic approach for detecting shock locations.

## 1 INTRODUCTION

Many natural phenomena are governed by conservation laws that take the form of hyperbolic partial differential equations (PDEs) (LeVeque, 2002). A challenging and ubiquitous feature of such PDEs is the formation of discontinuities, or shock waves, even when starting from smooth initial data. Detection and handling of shocks is the key challenge in simulation of such systems. When a shock appears, the differential form description (which relies on continuity) breaks down and one instead looks for "weak solutions" based on the corresponding integral conservation law (Osher, 1984). Any computational method needs dedicated tools for handling these sharp or discontinuous data, usually in the form of Riemann solvers and slope or flux limiters (Toro, 2013; LeVeque, 2002).

With the recent rise of machine learning technologies (deep learning (Goodfellow et al., 2016) in particular), there has been a lot of interest in using deep neural nets for scientific simulations, leading to the development of physics-informed neural networks (PINNs) (Raissi et al., 2019). PINNs are more difficult to train than the vanilla supervised learning problems (Krishnapriyan et al., 2021). PINNs loss is minimally composed of two parts: (a) A supervised loss at boundary(+initial) conditions and (b) Differential equation loss (PINN loss). Because the PINN loss relies on differentiation, continuously differentiable activation functions are preferred over the non-smooth ones that are popular in traditional supervised learning. This also suggests that PINNs might have difficulty solving PDEs with discontinuous solutions. Indeed it has been shown (Abreu & Florindo, 2021; Mojgani et al., 2023; Fuks & Tchelepi, 2020; Patel et al., 2022; Mao et al., 2020) that standard PINNs fail to capture shocks or fail to converge even for perhaps the simplest 1D hyperbolic conservation law, the inviscid Burgers equation:

$$\frac{\partial u}{\partial t} + \frac{1}{2}\left(u^2\right)_x = 0. \tag{1}$$

The application of PINNs to this problem has been studied in many papers and various strategies have been proposed to improve their performance (Mojgani et al., 2023; Abreu & Florindo, 2021; Patel et al., 2022). Similar issues have been observed in the application of PINNs to other hyperbolic systems, like convection-dominated flow (Krishnapriyan et al., 2021), the Buckley-Leverett equation (Fuks & Tchelepi, 2020; Abreu & Florindo, 2021), and compressible gas dynamics (Patel et al., 2022; Mao et al., 2020). A number of case studies with proposed solutions are listed in (Mojgani

---

[*]https://numerics.kaust.edu.sa/index.html

et al., 2023, Table 1). One of the popular remedies, which was also common in the early days of grid-based numerical solvers, is to add a very small (artificial) viscosity to the PDE (Fuks & Tchelepi, 2020; Abreu & Florindo, 2021; Patel et al., 2022). There exist a wide range of proposed solutions which vary in level of formulation or implementation ranging from reformulation of the problem to be optimized to adaptive sampling and from architectural considerations to enforcing additional constraints in the loss function.

The very definition of weak (discontinuous) solutions of hyperbolic PDEs is based on an integral conservation law, so it is natural to consider enforcing the integral form of the conservation law in order to capture discontinuities. Here we explore this idea by proposing a loss function based on the integral form, and we find that this significantly enhances the training of PINNs. In order to isolate and study the effect of this new approach to the loss function, we do not invoke any of the other strategies that have been proposed for dealing with discontinuities; many of those could be applied in addition and might lead to further improvements.

We note that our work is similar in spirit to that of Patel et al. (2022), as both works are based on the integral form of the conservation law. But in realization the approaches are very different, as Patel et al. (2022) use discretization of spacetime with a loss function which uses a numerical integration method. As will be clear from next section, we do not require these. Also, we use NN to model integral of the solution instead of solution itself and we do not need any domain-decomposition, making our approach conceptually different from other conservation-based PINNs (Patel et al., 2022; Jagtap et al., 2020; Hansen et al., 2023). In next section we outline the idea and then present case studies in further sections.

## 2 TRADITIONAL PINNS AND SHOCK-CAPTURING IPINNS

We consider systems of hyperbolic conservation laws of form

$$\frac{\partial U}{\partial t} + F(U)_x = 0 \tag{2}$$

$U(x,t) : \mathbb{R} \times \mathbb{R} \to \mathbb{R}^m$ is a set of conserved quantites (e.g. mass, momentum, energy) and $F$ is the corresponding flux function. For example, in the inviscid Burgers equation (1) above, $F(u) = \frac{1}{2}u^2$. The traditional PINNs approach is to apply a neural network ansatz for $U$ i.e.

$$\mathcal{N}_\theta(t,x) \approx U(t,x) \tag{3}$$

where $\mathcal{N}_\theta$ is a neural network with weights $\theta$. A Physics informed loss function is imposed on $\mathcal{N}_\theta$:

$$\mathcal{L}_{\mathcal{PI}} = \frac{1}{n_d} \sum_{j=1}^{n_d} \left( \frac{\partial \mathcal{N}_\theta(t_j, x_j)}{\partial t} + F(\mathcal{N}_\theta(t_j, x_j))_x \right)^2 \tag{4}$$

in addition to supervised loss on boundary/initial points (which we collectively denote by using subscript $b$ for boundary).

$$\mathcal{L}_{\mathcal{B}} = \frac{1}{n_b} \sum_{j=1}^{n_b} \left( \mathcal{N}_\theta(t_j, x_j) - U_j \right)^2 \tag{5}$$

where $U_j \coloneqq U(t_j, x_j)$ are values of function that we know from boundary/initial conditions. The total loss is:

$$\mathcal{L} = \mathcal{L}_{\mathcal{PI}} + \mathcal{L}_{\mathcal{B}} \tag{6}$$

Henceforth we will refer to the approach just described as *traditional PINN*. As discussed already, many improvements to traditional PINNs have been proposed already for hyperbolic PDEs, such as weighted loss or additional losses for improving conservation properties. But here we focus on evaluating a specific new formulation of the loss function.

Weak (i.e. discontinuous) solutions of the differential conservation law equation 2 must satisfy the integral conservation law (obtained by integrating equation 2 over an arbitrary spatial interval)

$$\frac{\partial}{\partial t} \int_{x_a}^{x_b} U(t,x) = F(U)|_{x_a} - F(U)|_{x_b}. \tag{7}$$

This relation holds for arbitrary intervals $[x_a, x_b]$. We leverage this fact by letting neural net $\mathcal{N}_\theta$ represent the integral of $U$ rather than $U$ itself:

$$\mathcal{N}_\theta(t, x) \approx \int_l^x U(t, x')(dx') \tag{8}$$

where $l$ is some reference point and can be conveniently taken as the left or right boundary of the spatial domain. Then the IPINN loss becomes:

$$\mathcal{L}_{\mathcal{IP}} = \frac{1}{n_d} \sum_{j=1}^{n_d} \left( \frac{\partial \mathcal{N}_\theta(t_j, x_{jb}) - \mathcal{N}_\theta(t_j, x_{ja})}{\partial t} - \left( F((\mathcal{N}_\theta)_x)|_{x_a} - F((\mathcal{N}_\theta)_x)|_{x_b} \right) \right)^2 \tag{9}$$

where $(\mathcal{N}_\theta)_x = \frac{\partial \mathcal{N}_\theta}{\partial x}$ and each training sample $j$ has three components $x_b, x_a, t$ i.e. each training point is composed of two different points from space with a time coordinate. So instead of a point-wise equation this is effectively a conservation equation over randomly-chosen intervals. The values of $U$ that are required for calculating fluxes at end points are calculated by taking the $x$-derivative of neural network. This is combined with the standard boundary loss function equation 5 to give a total loss function $\mathcal{L} = \mathcal{L}_{\mathcal{IP}} + \mathcal{L}_\mathcal{B}$.

## 2.1 DETECTING SHOCK LOCATIONS

A key challenge for hyperbolic PDEs with shocks is to detect the location of shocks. Even in the case where one is not looking for the neural-net based solution over the entire domain, it would be useful to have a (neural-net) model that informs us about the location of shocks in spacetime. This could further facilitate dynamic domain decomposition for either classical numerical solvers (similar to Beck et al. (2020)) or further training of domain-wise/domain-decomposition based neural networks (Shukla et al., 2021).

Here we look at the prospect of detecting shock locations based on the IPINN that we have trained. For simplicity, we will work with the example of 1-d Burgers of subsection 3.1. We notice that there are two indicators that can be used to detect the location of shocks. Both exploit the fact that there are equalities/inequalities that are satisfied in region of smoothly varying solutions but break down at shocks. The first indicator is based on the entropy inequality. Hyperbolic conservation law systems typically possess a so-called entropy-flux pair (LeVeque, 2002) $(\eta, q)$ such that

$$\mu(x, t) := \eta_t + q_x \tag{10}$$

vanishes except in the vicinity of a shock, where this quantity can be negative. For Burgers equation, one such entropy-flux pair is (Patel et al., 2022):

$$\eta \equiv u^2 \qquad\qquad q \equiv \frac{2}{3}u^3. \tag{11}$$

Given a trained IPINN, we can use the set of points where $|\mu|$ exceeds some threshold as an indication of where shocks may appear.

Another indicator of shock location is based simply on violation of the differential equation (2), and we discuss this in appendix A.1.

## 3 TEST CASES

In this section, we present experiments and comparison between IPINNs and traditional PINNs. We use solutions from the finite volume (FV) solver Clawpack (Clawpack Development Team, 2020) as reference solutions. Besides the examples in this section, we give an additional example in appendix A.2.

## 3.1 INVISCID BURGERS

We start with the inviscid Burgers equation (1), with smooth initial data:

$$u(0, x) = \begin{cases} 2\sin(3x) + \cos(2x) - 1.0, & \text{if } -\pi < x < \pi \\ 0, & \text{otherwise} \end{cases}$$

where $x \in [-10, 10]$, $t \in [0, 6]$ with constant (homogeneous Dirichlet) boundary conditions at $x = \pm 10$. Multiple shocks form after a short time. This example is even more challenging than, say, a Riemann problem, because of the transition from a smooth solution to a discontinuous one. Results are shown in Figure 1, where it can be seen that a traditional PINN completely fails to capture the shocks, while the IPINN captures both the amplitude and location of each moving shock fairly accurately, albeit still with some oscillations near two of the shocks. We also plot in pink the locations where $|\mu| > 1/2$ to show where this would indicate the presence of shocks, and we see that it coincides almost perfectly with the actual shock locations.

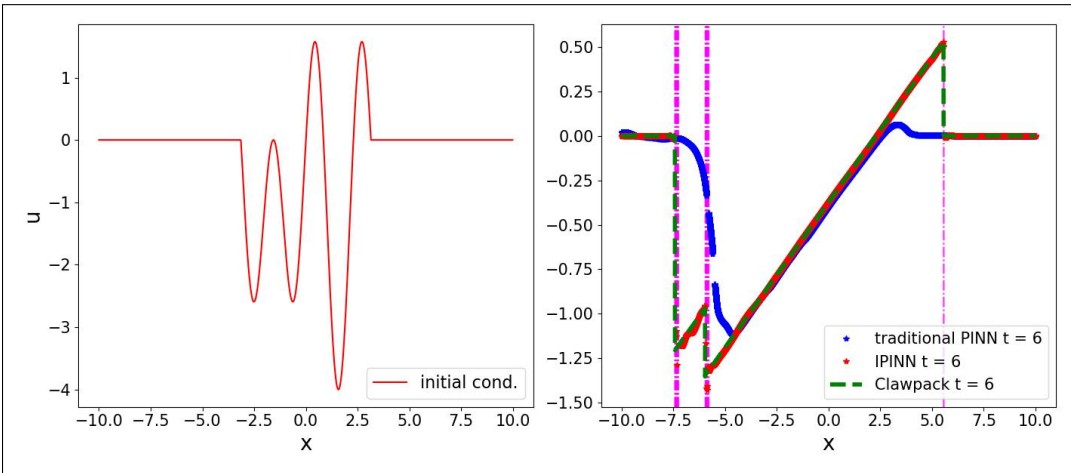

Figure 1: Comparing traditional PINNs with IPINNs for inviscid Burgers. **Left panel**: Continuous Initial data. **Right panel**: Solution at final time. The reference solution is computed using Clawpack. Traditional PINNs without any extra regularization fail to capture the moving shocks. IPINNs are able to handle these moving effects without any extra regularization. **Shock Locations**: Vertical magenta lines show where $\mu > 0.5$, suggesting the presence of a shock. These regions track the shocks with high accuracy.

### 3.2 SHALLOW WATER EQUATIONS (SWE)

The shallow water equations

$$h_t + (hu)_x = 0 \tag{12a}$$

$$(hu)_t + (hu^2 + \frac{1}{2}gh^2)_x = 0 \tag{12b}$$

are used to model water waves whose wavelength is long relative to the overall water depth, such as tsunamis in the deep ocean (Lannes, 2013). Here $h$ is the water depth and $u$ is the horizontal water velocity, while $g$ represents the force of gravity. We solve the Riemann problem with initial data

$$h(0, x) = 1 \qquad\qquad u(0, x) = \begin{cases} 1, & \text{if } x \leq 0.0 \\ -1, & \text{if } x > 0.0 \end{cases} \tag{13}$$

The solution consists of two outward-going shocks, and computed solutions are shown in Figure 2.

An entropy-flux pair for (12) is (Ketcheson & Quezada de Luna, 2022):

$$\eta \equiv \frac{1}{2}gh^2 + \frac{1}{2}hu^2, \qquad\qquad q \equiv \eta u. \tag{14}$$

We show the final shock locations, as indicated by entropy residual, in the lower-right plot of Figure 2. In this figure we plot $|\mu|$ (where now we take the appropriate entropy function and corresponding flux for the SWEs) for the two types of PINNs that we consider along with the profile of $h$ from Clawpack to show the discontinuity location as calculated by FV methods in Clawpack. We see again that IPINNs accurately capture the moving shock locations.

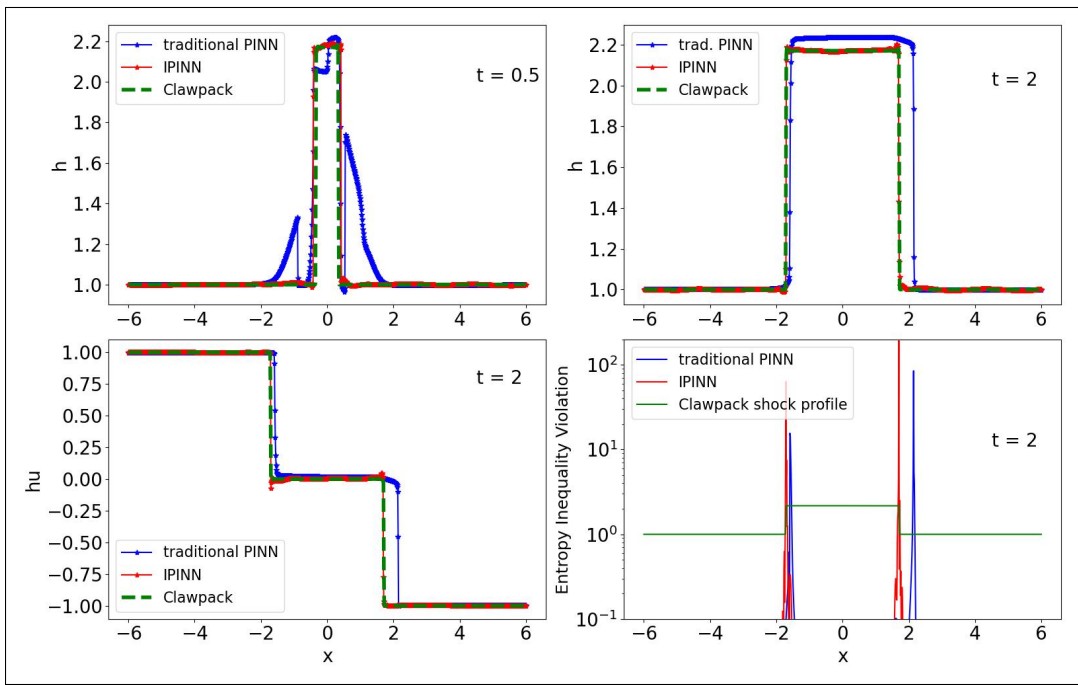

Figure 2: SWE evolved profiles. **Top panel**: Height ($h$) at two different times: ($t = 0.5$, left) ($t = 2.0$, right). Initial profile was uniform but then it breaks into these two moving fronts. **Bottom-left panel**: Momentum $hu$ at final time. $hu$ initially had a single discontinuity that later turns into two propagating fronts. Traditional PINN not only misses the final shock location, but also gives a completely incorrect solution at early times. In contrast, IPINNs significantly alleviate these shortcomings. **Bottom-right**: Entropy condition (11) violation with entropy-flux pair (14).

## 4    CONCLUSION, CHALLENGES AND FURTHER DEVELOPMENT

We have shown that putting the neural ansatz on the integral of the solution and using it in conservation form for PINN training enhances the performance of PINNs in learning solutions with moving shocks. In doing this we have not used any extra regularizations or modifications that have been proposed in literature. Those could be applied in combination with the IPINN approach to obtain possible further improvements. We note that IPINNs require extra autograd operations and can be slightly slower than traditional PINN implementations, but this is often also the case with other proposed modifications of PINNs.

We are currently working on generalizing this approach to 2D and 3D problems, as well as performing a cost-benefit analysis in terms of extra computational costs when scaling to systems of equations in 2D or 3D. The preliminary results in this work suggest that the proposed formalism might be more useful in scenarios where one expects to get strong moving shocks.

ACKNOWLEDGMENTS

The work was funded by King Abdullah University of Science and Technology.

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

## A  APPENDIX

### A.1  DIFFERENTIAL EQUATION VIOLATION FOR DETECTING SHOCK LOCATIONS

The differential form 2 of conservation laws break down where shocks are formed, so we can use "extreme peaks" of this differential form violation as an indicator of shock location. We define differential form violation ($DFV$) as:

$$DFV \equiv MSE(\frac{\partial U}{\partial t} + F(U)_x, 0) \tag{15}$$

Note that this is just the traditional Physics-informed loss. For a converged model, this is supposed hold everywhere but at shocks. We plot this quantity in figure 3 for the inviscid Burgers case at time corresponding to figure 1.

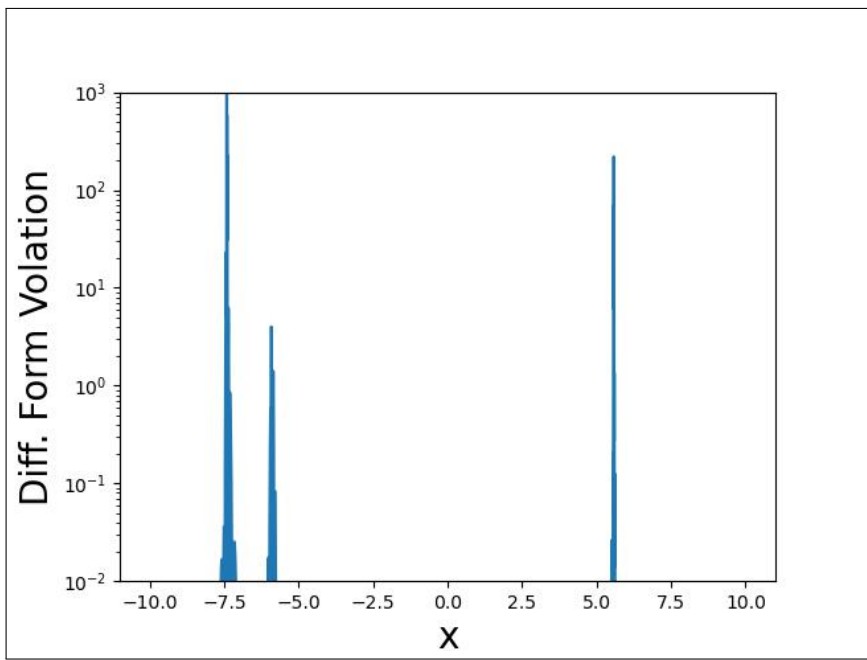

Figure 3: Differential form violation serves as an indicator of the shocks. Note that the y-axis is on log-scale and the sudden peaking of this violation corresponds to the discontinuities of figure 1.

### A.2  BUCKLEY-LEVERETT

Buckley-Leverett equation models two-phase transport problem with two fluids (Fuks & Tchelepi, 2020):

$$\frac{\partial u}{\partial} + \frac{\partial f(u)}{\partial x} = 0 \tag{16}$$

Here we consider the challenging case of non-convex flux function:

$$f(u) = \frac{u^2}{u^2 + (1 - u)^2} \tag{17}$$

with piecewise constant initial condition of $u(0, x) = 1$ for $x \leq 0$ and $u(0, x) = 0$ otherwise. One particular feature of this case that makes it complicated is the fact that the solution consists of a piecewise constant region, then a continuous rarefaction and a moving discontinuity. In the cases of Riemann problems where the evolution transforms piecewise constant initial data into piecewise constant data at any later time, one might use piecewise defined activation functions like relu and get very good results, but this would certainly fail for cases like Buckley-Leverett. Our experiments with Buckley-Leverett show that it is indeed tough to handle this complexity with simple PINNs, but the proposed IPINNs give much better results than traditional PINNs as shown in figure 4.

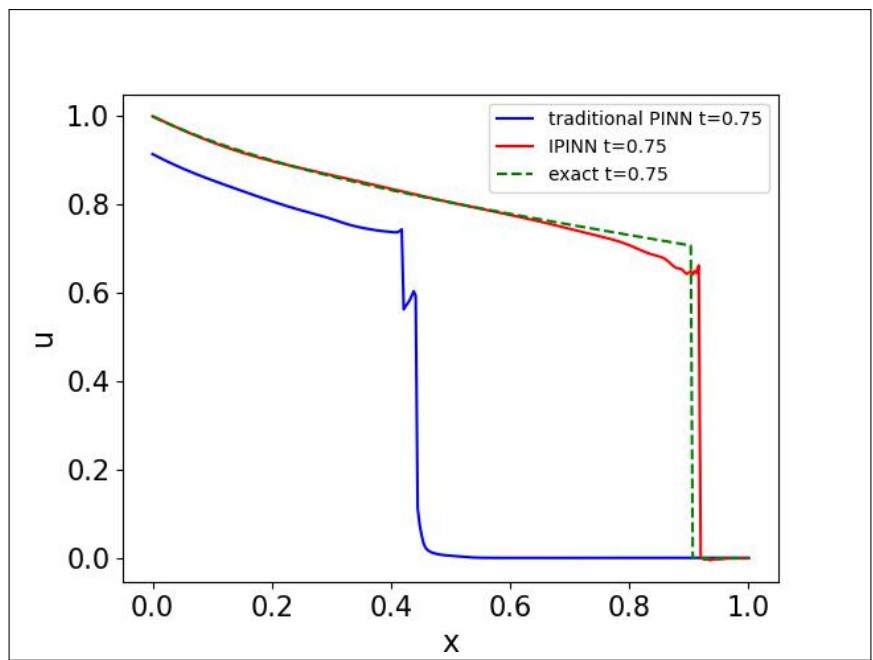

Figure 4: Buckley-Leverett: Final profile as predicted by two PINNs.

