# OpenReview forum: "INTEGRAL PINNS FOR HYPERBOLIC CONSERVATION LAWS"
_ICLR.cc/2024/Workshop/AI4DiffEqtnsInSci — AI4DiffEqtnsInSci @ ICLR 2024 Poster_

### Official Review · Reviewer_15Rz · 2024-02-14
**INTEGRAL PINNS FOR HYPERBOLIC CONSERVATION LAWS**

**Rating:** 6
**Confidence:** 4

**Review:**

The paper introduces Integral Physics-Informed Neural Networks (IPINNs) as an innovative approach to overcoming the limitations of traditional Physics-Informed Neural Networks (PINNs) in handling shock discontinuities in weak solutions of hyperbolic partial differential equations (PDEs). The novelty of IPINNs lies in their training methodology, which is based on the integral form of conservation laws. This method is advantageous because it remains valid across both continuous and discontinuous solution points, thereby facilitating more accurate modeling of shocks in hyperbolic conservation law systems.
The critique centers on the perceived lack of novelty in the proposed Integral Physics-Informed Neural Networks (IPINNs), noting their similarity to previous methods developed by authors Patel and Jagtap. To constructively address this concern, it is crucial to highlight any distinct features, improvements, or specific applications of IPINNs that differentiate them from the earlier approaches. This includes detailing any methodological enhancements, computational efficiencies, or broader applicability to a range of hyperbolic PDEs. Direct comparison in terms of performance metrics, accuracy in shock capturing, and computational resources required could further elucidate the advancements IPINNs offer over the previous methods. Without clear differentiation or demonstrated superiority, the contribution of the paper to the field may indeed appear limited.

---

### Official Review · Reviewer_X3h7 · 2024-02-22
**Effective use of integral form in PINNS, clear accept**

**Rating:** 8
**Confidence:** 4

**Review:**

# Summary
This paper introduces integral physics-informed neural networks (IPINNs), which differ from traditional PINNs by training a neural network to model the integral of the solution rather than the solution itself. This is in contrast to other PINN papers that attempt to incorporate integral conservation laws by numerically integrating the sub-regions [1] or modelling flux on interfaces between sub-regions [2].

This approach is targeted to hyperbolic PDEs, where traditional PINNs are known to break down because the solution is discontinuous near "shock" points. This paper shows that the IPINNs capture the solution profile better than traditional PINNS on two challenging problems (Inviscid Burgers' and Shallow Water Equations).

# Review
Overall, this work warrants acceptance to the workshop. The modification to PINNs to incorporate the integral form is elegant in its simplicity. More importantly, the authors demonstrate that their method is effective on challenging hyperbolic PDEs where PINNs are not.

## Originality
This is not the first work to integrate the integral form of conservation laws into PINNs. However, this work differs from previous approaches by having the neural network learn the integral of the solution over the state domain, rather than the solution itself. The original solution is then recovered by auto-differentiation of the neural network. To my knowledge, this seems original, but there may have been something I have missed in the deluge of PINNs-related literature.

## Significance
Unlike many PINNs-related papers, this work is targeted at difficult, hyperbolic PDEs that pose challenges for both PINNs and traditional numerical methods. By taking advantage of knowledge of the PDE (hyperbolic with conservation law), the performance of PINNs can be greatly improved. This illustrates that traditional PINNs are not a "plug and play" approach that can solve any PDE with the same framework.

## Quality
The method is relatively straightforward as written. The authors change the loss to be on the integral of $U$ rather than $U$ itself. However, this makes a large difference in the performance of the network in empirical settings (Figures 1 and 2). The authors pick challenging examples that are a failure mode of traditional PINNs, and aim to solve an important downstream task (shock detection).

While not necessarily required for a workshop paper, I would expect in a full paper to see an empirical comparison to the referenced competing "conservative" PINNS approaches such as cvPINNs [1] and cPINNs [2]. My hunch is that these other methods will break down in the hyperbolic setting, but it would be good to validate that hunch empirically.

## Clarity
The method and paper are clear and easy to follow. However, Figure 1 (left panel) and Figure 2 are quite difficult to read. It is difficult to visually distinguish the solutions. I suggest breaking out each competing method into a separate facet that is compared with just the reference (Clawpack).

Also, I have a minor stylistic comment. It is confusing to have the neural network be referred to as $N$ and $N_\theta$. The use of capital "N" is typically reserved for a number (e.g. number of samples). This is especially confusing when in Equation 8 the lowercase $n_d$ refers to the dimension size.

[1] Patel, Ravi G., Indu Manickam, Nathaniel A. Trask, Mitchell A. Wood, Myoungkyu Lee, Ignacio Tomas, and Eric C. Cyr. “Thermodynamically Consistent Physics-Informed Neural Networks for Hyperbolic Systems.” Journal of Computational Physics 449 (January 15, 2022): 110754. https://doi.org/10.1016/j.jcp.2021.110754.

[2] Jagtap, Ameya D., Ehsan Kharazmi, and George Em Karniadakis. “Conservative Physics-Informed Neural Networks on Discrete Domains for Conservation Laws: Applications to Forward and Inverse Problems.” Computer Methods in Applied Mechanics and Engineering 365 (June 15, 2020): 113028. https://doi.org/10.1016/j.cma.2020.113028.

---

### Official Review · Reviewer_WRKx · 2024-02-26
**Good Paper, Accept**

**Rating:** 7
**Confidence:** 4

**Review:**

I congratulate the authors on getting this research to this stage.

The research is good quality and clearly describe the problem and propose the solution. The work presented is known to be original (in my knowledge) and marks an significant improvement to use IPINNs for hyperbolic conservation laws.

The text is written in clear manner and easily understandable. There are some minor inconsistencies in the text , eg. neural net and nets are used interchangeably, Line 4 use the word "such" and rephrasing is needed here. This is not an exhaustive list and therefore I request authors to do a quick proof reading.

---

### Meta-Review · Area_Chair_aRbF · 2024-02-22

**Recommendation:** Accept (Poster)

**Metareview:**

The reviewers highlight the importance of the proposed method empirically for solving hyperbolic conservation laws with shocks, which can be a failure mode of the original PINN method. I vote for acceptance but agree with the other works, e.g., cPINNs, cvPINNs and ProbConserv [Hansen et. al, "Learning Physical Models that Can Respect Conservation Laws", ICML 2023] that also proposes to use the integral form of the conservation law, should be compared to or at least referenced. Note that there is a minor typo in Eqn. 7, where $x_b$ is missing.

---

### Decision · Program_Chairs · 2024-02-28

Accept (Poster)